# Prevention of HBV Reactivation in Hemato-Oncologic Setting during COVID-19

**DOI:** 10.3390/pathogens11050567

**Published:** 2022-05-11

**Authors:** Caterina Sagnelli, Antonello Sica, Massimiliano Creta, Alessandra Borsetti, Massimo Ciccozzi, Evangelista Sagnelli

**Affiliations:** 1Department of Mental Health and Public Medicine, Section of Infectious Diseases, University of Campania “Luigi Vanvitelli”, 80131 Naples, Italy; evangelista.sagnelli@unicampania.it; 2Department of Precision Medicine, University of Campania “Luigi Vanvitelli”, 80131 Naples, Italy; antonello.sica@fastwebnet.it; 3Department of Neurosciences, Reproductive Sciences and Odontostomatology, University of Naples Federico II, 80131 Naples, Italy; massimiliano.creta@unina.it; 4National HIV/AIDS Research Center, Istituto Superiore di Sanità, 00161 Rome, Italy; alessandra.borsetti@iss.it; 5Medical Statistics and Molecular Epidemiology Unit, Campus Bio-Medico University, 00128 Rome, Italy; m.ciccozzi@unicampus.it

**Keywords:** COVID-19, SARS-CoV-2, hepatitis B virus, reactivation, chemotherapy, immunosuppression, prevention

## Abstract

Onco-hematologic patients are highly susceptible to SARS-CoV-2 infection and, once infected, frequently develop COVID-19 due to the immunosuppression caused by tumor growth, chemotherapy and immunosuppressive therapy. In addition, COVID-19 has also been recognized as a further cause of HBV reactivation, since its treatment includes the administration of corticosteroids and some immunosuppressive drugs. Consequently, onco-hematologic patients should undergo SARS-CoV-2 vaccination and comply with the rules imposed by lockdowns or other forms of social distancing. Furthermore, onco-hematologic facilities should be adapted to new needs and provided with numerically adequate health personnel vaccinated against SARS-CoV-2 infection. Onco-hematologic patients, both HBsAg-positive and HBsAg-negative/HBcAb-positive, may develop HBV reactivation, made possible by the support of the covalently closed circular DNA (cccDNA) persisting in the hepatocytic nuclei of patients with an ongoing or past HBV infection. This occurrence must be prevented by administering high genetic barrier HBV nucleo(t)side analogues before and throughout the antineoplastic treatment, and then during a long-term post-treatment follow up. The prevention of HBV reactivation during the SARS-CoV-2 pandemic is the topic of this narrative review.

## 1. Introduction

The severe acute respiratory syndrome coronavirus 2 (SARS-CoV-2) pandemic has spread worldwide with five successive waves, causing serious damages to healthcare systems and the economy of nations worldwide. Many clinical facilities and health personnel have been assigned to care for patients with coronavirus disease 2019 (COVID-19), reducing the level of care for patients with other diseases, including those with blood malignancies.

The reactivation of hepatitis B virus is a frequent event in onco-hematologic patients, both hepatitis B surface antigen (HBsAg)-positive and HBsAg-negative/hepatitis B core antibody (HBcAb)-positive ones, since the covalently closed circular DNA (cccDNA) remains a small microsome in the nucleus of infected hepatocytes even after patients have recovered, acting as a template for HBV-DNA synthesis and able to induce HBV reactivation. The immunosuppression induced by the growth of the tumor and by antineoplastic treatments is the main cause of this reactivation.

COVID-19 has also been recognized as a further cause of HBV reactivation, since its treatment includes the administration of high-dose corticosteroids and/or some immunosuppressive drugs.

HBV reactivation should be prevented by administering high genetic barrier HBV nucleus(t)side analogues as part of a careful clinical and laboratory long-term follow-up.

This narrative review analyzes the impact of COVID-19 on onco-hematologic patients, with reference to HBV reactivation in HBsAg-positive and HBsAg-negative/HBcAb-positive patients.

The article is particularly aimed at students at specialization schools and doctors working in the sectors of onco-hematology, infectious diseases and hepatology, and to those who in any capacity are involved in the care of patients with COVID-19.

## 2. Pathogenic Mechanisms of SARS-CoV-2 Infection and COVID-19

Identified at the end of 2019, SARS-CoV-2 assumed the characteristics of a pandemic in just a few months [1,2,3,4]. Two-thirds of patients are asymptomatic or have mild symptoms, and in the remaining third, this infection progresses to a symptomatic form called COVID-19, presenting in varying degrees of severity depending on the host’s immune response to the virus. The essential step to infect humans is the binding of the SARS-CoV-2 spike proteins to angiotensin-converting enzyme 2 (ACE2) receptors, present in human cells. It follows a stimulation of the host’s immune system, which can result in a cytokine storm and in a systemic inflammatory syndrome responsible for a severe and sometime fatal illness [3,4,5,6,7,8,9,10,11]. COVID-19 is mainly characterized by interstitial pneumonia, frequently severe and sometimes fatal, but other organs and systems could be affected because ACE2 receptors are widespread in human tissues.

Compared to younger subjects, those over 60 more frequently develop a severe clinical course [12,13,14,15,16,17,18,19,20,21,22,23] and more frequently develop respiratory failure, heart failure, renal failure, sepsis or multi-organ failure. An excessive inflammatory response can induce a cascade of reactions, leading to blood clotting with an increased risk of intravascular clots and pulmonary embolism [24]. There is also nervous system involvement, evidenced by an impaired cognitive attention and memory capacity [25,26,27,28,29].

## 3. Pathogenic Mechanism of HBV Infection and HBV Reactivation

HBV infection induces an acute hepatic necroinflammation (acute hepatitis B (AHB)) of varying extent and a subsequent hepatocyte proliferation. In patients with a “normal” immunological balance, AHB is followed by the resolution of HBV infection, whereas it progresses to chronicity in those with an impaired immune response to the virus. HBV becomes chronic in 90% of newborn babies born of HBeAg-positive mothers, in 80–90% of infants infected during the first year of life [30], in 20–50% of children under six years old, in 6% of those aged 6–15 years and in 1–5% of older patients [31].

The clinical course of HBV chronic infection varies from an asymptomatic silent progression over decades, more frequent in children than in adults, to a rapid evolution to chronic hepatitis, liver cirrhosis and hepatocellular carcinoma (HCC), which is more frequent in adults [32,33].

Several factors may influence this progression: host-related (immune response) issues, virus-related (HBV genotype, length of HBV infection, HBV viral load, coinfection with hepatitis delta virus, hepatitis C virus and human immunodeficiency virus) issues, comorbidities, alcohol abuse and immunosuppression [34,35,36,37].

HBV chronic infection presents a dynamic evolution though four main phases, not necessarily consecutive: an “Immune tolerant phase” (mild/absent necroinflammation, normal/low aminotransferases serum values, high HBV load, HBeAg positivity and slow or absent fibrosis); an “immune reactive phase” (CHB with liver necroinflammation, increased or fluctuating aminotransferases serum values, a low/intermediate HBV load, HBeAg positivity and possible transition to cirrhosis); an “inactive HBV carrier state” (with low/absent necroinflammation, low/normal aminotransferase serum values, a low or undetectable serum HBV load and seroconversion to anti-HBe); “HBeAg-negative CHB” (with mild to severe hepatic necroinflammation, fluctuating aminotransferases serum values and detectable HBV-DNA, linked to an e-minus HBV variant unable to express HBeAg) [38].

Most children with HBV chronic infection remain asymptomatic HBsAg carriers for decades, and only a minority of them progress to CHB. Instead, a high percentage of adults with HBV chronic infection presents with CHB, which may over time progress to liver cirrhosis, a condition that favors the development of HCC.

Hepatitis B virus is a non-cytopathic virus with a diameter of 42 nm and looks like a spherical particle at electron microscopy, with a core composed of the hepatitis core antigen (HBcAg), a double-stranded circular DNA and reverse DNA polymerase/transcriptase (Pol/RT), surrounded by an envelope carrying some surface antigens named HBsAg, pre-S1 and pre-S2 [39,40]. The genome consists of 3200 base pairs in four overlapping open reading frames encoding surface proteins, the core protein, a DNA-dependent polymerase/transcriptase enzyme and a non-structural X protein [40,41,42,43].

HBV infects human hepatocytes and viral replication proceeds through a reverse transcription of a pre-genomic RNA and the subsequent synthesis of viral DNA [44,45,46]. HBV covalently closed circular DNA (cccDNA) presents in the nuclei of infected hepatocytes in the form of a microsome, directing the transcription of viral RNA [30]. This microsome is quite stable in infected hepatocytes, with its level replenished and amplified by the replicating HBV-DNA [47].

A couple of weeks after infection, the elimination of HBV already begins through non-cytopathic mechanisms, mostly mediated by antiviral cytokines (tumor necrosis factor α, interferon α and interferon β) produced by the cells of innate and adaptive immune responses [48,49].

Then, the activation of the cytolytic immune response causes apoptosis and hepatocyte necrosis, resulting in the onset of the acute phase of the disease. The main mechanism causing both liver damage and virus control is based on the recognition of HBV-infected hepatocytes by virus-specific CD8 cytotoxic T cells via HBV core-derived peptides exhibited by human lymphocyte class I (HLA-I) antigens. In addition, cytotoxic T cells in the liver activate antigen-non-specific inflammatory cells to secrete cytokines, initiating a cascade of immunological events inducing necroinflammation [31,43,50,51,52,53,54,55,56].

HBV clearance depends on a vigorous multi-specific CD4+ T and CD8+ T response [57]. Both CD8+ T cells with their cytotoxic effector function and CD4+ T cells are critical for achieving viral clearance, as CD4+ T cells are required both for the development of optimal CD8+ effector T cells and for the generation and maintenance of a function memory of CD8+ T lymphocytes [58,59].

A prolonged vigorous multi-specific CD4+T and CD8+ T response is crucial to clear HBV infection in AHB, while its exhaustion in the phase of HBV elimination determines the persistence of the virus and of an inflammatory response ineffective for virus clearance [50,57,60,61,62,63,64].

The persistence of cccDNA in a latent state in the hepatocytic nuclei of both patients with ongoing chronic HBV infection and subjects with past HBV infection is the substrate for HBV reactivation [65]. The cccDNA withstands nucleo(t)side analogues, drugs widely used to suppress HBV replication, which, even if administered for years, may only partially reduce cccDNA activity [66,67].

Corticosteroids, chemotherapy and immunosuppressive therapy are the most frequent causative agents of HBV reactivation, but the discontinuation of HBV suppressive therapy with high genetic barrier nucleo(t)side analogues, entecavir (ETV), tenofovir disoproxil fumarate (TDF) or tenofovir alafenamide fumarate (TAF) is acquiring an increasingly well-defined role in this area [68,69,70,71,72].

The reactivation of HBV replication implies the resumption of the immune response to the virus and a consequent liver necroinflammation, frequently not extensive, but in some cases long-lasting and responsible for life-threatening subacute hepatitis and in others for massive and acute liver failure [72,73].

## 4. HBV Reactivation in Patients with Hemato-Oncologic Malignancies and Its Prevention

Before starting any antineoplastic treatment, hemato-oncologic patients should be tested for serum HBsAg and HBsAb, and the positive patients also for the serum HBV-DNA load. Being at a high risk of HBV reactivation, HBsAg-positive patients with cancer have to undergo prophylaxis with a high genetic barrier nucleus(t)side analogue (ETV, TDF or TAF) to be started one–two weeks before antineoplastic therapy [73], whereas pre-emptive therapy with the same nucleus(t)side analogues is considered sufficient for HBsAg-negative/HBcAb-positive patients who are at a lower risk of HBV reactivation; an exception should be made for the HBsAg-negative/HBcAb-positive patients intending to receive treatment with anti-CD20 antibodies or to undergo stem cell transplantation (SCT), who would need nucleus(t)side analogues in prophylaxis [74,75,76].

Of note, pre-emptive therapy with high genetic barrier nucleo(t)side analogues implies the determination of HBV-DNA serum levels every 1–3 months during antineoplastic treatment and the subsequent post-treatment follow-up. Once started, nucleo(t)side analogue prophylaxis should be continued throughout the antineoplastic therapy and a 12-month post-treatment follow-up, which should be extended for a further 6 months for patients treated with anti-CD20 antibodies or SCT [77,78,79].

The published studies on HBV reactivation in cancer patients concern patients unprotected by high genetic barrier nucleo(t)side analogues. It has been estimated that anticancer therapy induces HBV reactivation in 41–53% of HBsAg-positive subjects and in 8–18% of those that are HBsAg-negative/HBcAb-positive [80,81], with rituximab, alemtuzumab, anthracyclines, fludarabine and high-dose corticosteroids being the drugs most frequently involved. Some authors have highlighted that the type of malignancy also plays a role in the development of HBV reactivation [82,83,84]. In a retrospective study on 156 HBsAg-positive neoplastic patients undergoing chemotherapy, the incidence of severe HBV exacerbation was 25% in those with hematological malignancies, and 4.3% in those with solid tumors; of note, this rate was 40% in patients receiving rituximab-based chemotherapy and 4.1% in those treated with rituximab-free chemotherapy [85].

Numerous studies concern the reactivation of HBV in HBsAg-negative/HBcAb-positive onco-hematologic patients not protected by a high genetic barrier nucleus(t)side analogue. Wu et al. [86] reported a 5.7% incidence rate in leukemia patients receiving chemotherapy and a 2.2% in those who underwent SCT [86].

Hui et al. [87] found a 3.3% HBV reactivation in 244 patients with malignant lymphoma, with a higher reactivation rate in those treated with rituximab plus corticosteroids.

In a prospective study performed by Yeo et al. [88], in patients with diffuse large B-cell lymphoma (DLBCL), HBV reactivation occurred in 23.8% of the 21 receiving R-CHOP and in none of the 25 treated with CHOP.

In 150 patients with lymphoma receiving rituximab-based chemotherapy, Hsu et al. [89] found an incidence of HBV reactivation of 10.4 per person per year and the development of a hepatic flare in 6.4% of cases [89].

Matsui et al. [90] followed up with 59 patients with lymphoma under R-CHOP treatment, and found HBV reactivation in 6.8% of cases.

Ji et al. [91] reported HBV reactivation in 1 patient out of 43 with DLBCL treated with R-CHOP. The different frequency of HBV reactivation in HBsAg-negative/HBcAb-positive patients with onco-hematologic malignancies across the above-mentioned studies reflects differences in selection criteria, duration and the quality of the follow-up, type of onco-hematologic malignancy, degree of patients’ immunosuppression and type of antineoplastic treatment. Taken together, however, these studies indicate that HBV reactivation affects 2–10% of HBsAg-negative/HBcAb-positive patients, the highest rates occurring in those treated with rituximab-based chemotherapy.

## 5. SARS-CoV-2 Infection and COVID-19 in Onco-Hematologic Patients

Cancer patients are at a high risk of developing severe COVID-19 once infected with SARS-CoV-2 [92,93,94,95,96,97]. Due to their low or absent host immune response, this virus is free to replicate and spread in patients with neutropenia, aplasia, bone marrow hypoplasia or neoplastic bone marrow disease, with massive direct cell damage; serous dangers are also exerted in patients with indolent chronic hematological diseases, in those who have undergone a bone marrow transplant and in those receiving or having recently received immunosuppressive therapy or chemotherapy [18,96,98,99,100,101]. In contrast, SARS-CoV-2 infection can induce an immunological hyperactivation and cytokine storm with consequent indirect tissue damage in patients with onco-hematological diseases involving antigen-presenting cells, T lymphocytes, NK or histiocytes [11,18,19,98,99,100,101]. This can also occur in patients with multiple myeloma on immunomodulatory therapy and in those with chronic myeloproliferative diseases [18,19,98,99,100,101].

The management of cancer patients during the SARS-CoV-2 pandemic has been complex. The real need of patients to access health facilities should be carefully assessed through an online triage to distinguish between deferrable and non-deferrable hospitalization needs. During a pandemic wave, hospitalization could be postponed by 12 weeks for patients waiting for a non-urgent transplantation and for those with stabilized diseases [102,103,104,105]. Patients on the waiting list for allogeneic transplantation should start conditioned chemotherapy only after the arrival of the donor cells and cryopreservation.

It has been recommended by several scientific societies (the European Society for Medical Oncology, the American Society of Clinical Oncology, the National Comprehensive Care Network, the Commission on Cancer, the National Institute for Health and Care Excellence in the UK and the American Society of Radiation Oncology) that a diagnostic molecular nasopharyngeal swab should be performed to detect SARS-CoV-2 RNA by PCR in all patients with cancer who have signs or symptoms of an influenza-like illness (ILI), in those who have been in contact with individuals with SARS-CoV-2 infection and in those who have been in hyperendemic areas; the positive ones should be admitted to dedicated care facilities until the RNA test is negative in at least two consecutive swabs [106,107,108,109,110].

Patients with hematological cancer should be advised to minimize hospital visits to prevent SARS-CoV-2 infection and to remain in telephone and audiovisual contact with the medical staff in care. The use of telemedicine can be of considerable support in the management of onco-hematological patients who can receive useful information on the behaviors to be followed, the modulation of therapy and any possible need they may have [106,110,111,112]. When necessary, onco-hematologic patients who have completed the SARS-CoV-2 vaccination course may be admitted with caution to healthcare facilities if they test negative with a SARS-CoV-2 RNA nasopharyngeal swab and have no ILI symptoms nor contact with SARS-infected subjects.

Most of the published studies on the efficacy of the COVID-19 vaccine report data obtained with the administration of two doses. Taken together, these studies demonstrate a good vaccine efficacy in the general population and lower efficacy in cancer patients [113,114,115,116]. In fact, Embi et al. [114] found that two doses of the COVID-19 mRNA vaccine prevented hospitalization in 90% of immunocompetent subjects, in 77% of immunocompromised patients and in 75% of cancer patients. In a retrospective, multi-center cohort study on 184,485 US cancer veterans, Wu et al. [115] evaluated COVID-19 mRNA vaccination as effective in approximately 60% of patients, ranging from 54% in patients on endocrine therapy or chemotherapy to 85% in those left untreated in the last 6 months [115]. Therefore, even if somewhat less effective than in normal subjects, COVID-19 vaccination is of great use for onco-hematologic patients. The need to vaccinate infection immunosuppressed and cancer patients against SARS-CoV-2 with a complete vaccination cycle (three vaccine doses) and with additional doses eventually required by the evolution of the pandemic and the duration of vaccine protection was clearly demonstrated in a UK study performed by Hippisley-Cox on a prospective cohort of 6,952,440 vaccinated subjects, of whom 5,150,310 received two doses of vaccine and the remaining ones a single dose. The study highlighted risk factors for death from COVID-19 to be moderate or high intensity chemotherapy (HR 3.63 and 4.3, respectively), stem cell transplantation within the last six months (HR 2.5), hematological cancer (HR 1.86) and cancer of the respiratory tract (HR 1.35) [116]. There was then the clear indication that oncologic patients should be given priority in COVID-19 vaccination over healthy citizens. This is true even today, since the administration of the fourth dose of RNA vaccine has begun in high-income countries with priority for the elderly and cancer patients.

However, there are concerns regarding the vaccination of numerous patients due to insufficient information, misinformation transmitted by social and mass media, the wavering decisions of some governments and patient fears of worsening the precarious equilibrium of the diseases. Evidence of these concerns was provided by Mejri et al. [117], who interviewed 329 Tunisian cancer patients; the vaccination acceptance rate was 50.5%, while 28.3% refused and 21.2% remained undecided and did not get vaccinated. The most frequent reasons for non-acceptance were the fear that the COVID-19 vaccine was not sufficiently effective and that it could negatively affect the course of the disease and the efficacy of treatments in place. This is a useful indication for physicians and psychologists to better target their work in convincing cancer patients that this vaccine is effective, safe and well-tolerated by almost everyone.

The reasons for vaccine hesitancy are heterogeneous and complex [118]. Three main motivations were identified by the SAGE group: contextual influences (e.g., socioeconomic and political), individual and social influences (e.g., social or personal experiences) and problems related to the vaccine and its administration (e.g., perceived benefits and risks and attitudes of health care providers) [119]. In June–August of 2020, there were conflicting views on the need for a COVID-19 vaccine, attributable to several factors that helped create vaccine hesitancy, including the perceived low risk of COVID-19 infection, vaccine-specific concerns and low adherence to COVID-19 information sources [120,121,122]. This hesitancy has also arisen among a significant number of physicians, and this has further increased hesitation about the COVID-19 vaccine in the general population. It is reasonable to hypothesize that vaccine hesitation could be lessened if it were administered by family doctors and doctors involved in the clinical and therapeutic follow-up of patients with long-lasting diseases, as it commonly occurs for onco-hematological patients. Trust in long-time respected doctors can increase patient confidence in the vaccine, and the practice of vaccination can increase doctors’ confidence in a vaccine whose benefits they have directly assessed.

## 6. HBV Reactivation in Patients with COVID-19

There is little information on HBV reactivation in patients with COVID-19 due to the short time elapsed since the start of the COVID-19 pandemic. Wu et al. [91] published the case of a 45-year-old male affected by HBsAg-positive chronic hepatitis treated with adefovir dipivoxil and ETV, who was hospitalized for COVID-19. He was HBsAg-positive and HBeAg/HBeAb-negative, his serum aminotransferases were normal and HBV-DNA was undetectable. The patient was treated with methylprednisolone and developed a moderate HBV reactivation identified by a moderate increase in aminotransferases serum values and by a moderate HBV-DNA load. TDF was added to therapy and HBV reactivation rapidly resolved. The nasopharyngeal molecular swab to detect SARS-CoV-2 RNA became permanently negative and the patient recovered from COVID-19 [123].

Aldhaleei et al. [124] published the case of a 35-year-old man hospitalized for unconsciousness, which developed after an episode of vomiting, and was found positive for SARS-CoV-2 infection. He was jaundiced with serum aminotransferases over 100 times the higher value of the normal, had elevated serum bilirubin and slightly reduced serum albumin levels; a brain CAT scan did not show pathological findings. He was HBsAg-positive, HBV-DNA-positive, HBcAb IgM-positive, HBeAg-negative and HBeAb-positive. No corticosteroids nor immunosuppressive drugs were administered except for supportive therapy and entecavir one mg per day. Liver disease gradually resolved, and the SARS-CoV-2 RNA test became negative. In the absence of information on HBV markers prior to admission and relying on HBeAb positivity, the authors hypothesized the diagnosis of severe HBV reactivation related to SARS-CoV-2 infection, but the possibility of an acute hepatitis B concomitant with SARS-CoV-2 infection could not be excluded.

In a prospective cohort study of 600 COVID-19 patients, 61 were found to be HBsAg-negative/HBcAb-positive and were followed up with to identify a possible HBV reactivation during corticosteroid and/or immunosuppressive therapy. Out of the 61, 38 received ETV prophylaxis (0.5 mg/day) and 23 remained untreated. HBV-DNA became detectable in 2 of 23 untreated patients and in none of 38 in the prophylaxis group, suggesting that entecavir was effective in preventing HBV reactivation [125].

The monoclonal antibody tocilizumab (TCZ) is often used in COVID-19 centers to neutralize the negative and life-threatening effects of the cytokines storm. Particularly evaluated in studies on rheumatoid arthritis, this drug has induced HBV reactivation in nearly 3% of treated patients [126]. There are no literature data on HBV reactivation in patients with COVID-19 treated with TCZ, nor on its prevention with high genetic barrier nucleo(t)side analogues, but, pending studies on the subject, we consider it prudent to associate these drugs with TCZ in patients at risk of reactivation.

## 7. Discussion

SARS-CoV-2 has been with us for more than two years, during which it has overwhelmed health facilities and destabilized the economic systems of all nations, causing nearly 500 million documented infections worldwide. SARS-CoV-2 is a virus with a great replicative capacity, able to mutate rapidly to form stable variants responsible for pandemic waves.

The harmful effect of SARS-CoV-2 on humans lies in its ability to stimulate a cytokine storm responsible for a systemic inflammatory syndrome, which leads to the development of COVID-19, a disease that has so far caused 6 million deaths worldwide. Many health facilities and staff originally assigned to the care of onco-hematologic patients were converted to the needs of COVID-19 patients, with a significant reduction in the quality of care for onco-hematologic patients, including a reduction in space, dedicated staff and activities for outpatients. This also resulted in an increased risk of HBV reactivation in onco-hematological patients with present or past HBV infection. HBV reactivation is a frequent event in onco-hematologic patients, due to the immunosuppression related to tumor proliferation, chemotherapy and immunosuppressive therapy that must be prevented, as it can trigger a series of immune-mediated reactions responsible for hepatic necroinflammation, sometimes subacute or massive, with a fatal outcome. Its prevention includes a pre-treatment screening to assess the HBV replicative status and the administration of a high genetic barrier nucleo(t)side analogue administered in prophylaxis or as pre-emptive therapy as needed, starting one–two weeks before the antineoplastic treatment and continuing throughout its duration and during the subsequent long-term follow-up.

ETV, TDF or TAF should be administered in prophylaxis in HBsAg-positive patients and HBsAg-negative/HBcAb-positive ones undergoing stem cell transplantation or rituximab-based chemotherapy. Pre-emptive treatment can only be chosen for HBsAg-negative/HBcAb-positive patients with no intention to be cured with anti-CD20 antibodies, nor planning to undergo stem cell transplantation; if that is the choice, however, it is essential to assess the HBV-DNA load every 1–3 months throughout treatments and the long-term post-treatment follow-up.

The optimal management of patients with onco-hematologic neoplasms, both HBsAg-positive and HBsAg-negative/HBcAb-positive, requires coordinated choices by a multidisciplinary team to establish the therapeutic strategies, methods and timing of checkpoints and tasks of each team member.

A full-course anti-SARS-CoV-2 vaccination is the most effective weapon against COVID-19, a very useful prophylactic measure for onco-hematologic patients in preventing the dangerous effects of COVID-19. However, approximately half of cancer patients have doubts about its efficacy and fear that it could upset the balance of their already precarious health conditions or negatively influence the effectiveness of antineoplastic therapy.

To dispel their doubts, doctors and psychologists in care have to provide their patients with simple and truthful information on the damage caused by the disease and on the efficacy, safety and tolerability of the vaccine.

## 8. Conclusions

HBV reactivation, a frequent event in HBsAg-positive and HBsAg-negative/anti-HBc-positive onco-hematologic patients, in some cases induces subacute or massive hepatic necroinflammation with severe clinical outcomes. Its prevention in this area includes the identification of HBsAg-positive and HBsAg-negative/anti-HBc-positive patients prior to the initiation of antineoplastic therapy and their protection with high genetic barrier nucleus(t)side analogues, administered in prophylaxis or as pre-emptive therapy as needed.

During the COVID-19 pandemic, onco-hematologic patients and the personnel assigned to their care must be protected by a full-course SARS-CoV-2 vaccination and the health facilities for their assistance must be adapted to the needs deriving from the pandemic.

## Data Availability

Not applicable.

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
