# Peer review of "Prevention of HBV Reactivation in Hemato-Oncologic Setting during COVID-19"

_pathogens, 2022, doi:10.3390/pathogens11050567_

Round 1

Reviewer 1 Report

In this manuscript titled “Prevention of HBV reactivation in hemato-oncologic setting during COVID-19”, the authors briefly went over the basics of COVID-19 and HBV infections, then introduced the potential HBV reactivation in hemato-oncologic patients, especially in the context of COVID-19 pandemic. Finally, the authors proposed that vaccination against SARS-CoV-2 is the most effective way to deal with such challenging clinical situation. Overall, this is a timely and interesting review. However, I feel that some more detailed information and more references are needed.

Major issues:

  1. Line 78-79, “There is also nervous system involvement, evidenced by impaired cognitive attention and memory capacity”, references are needed for this statement.

  1. Line 82, what is the meaning of “pathogenetic mechanism” of HBV infection here? If it means hepatitis, liver failure, cirrhosis, or cancer, it is directly related with hepatocyte death or overgrowth, rather than T cell activation. Please define the pathology consequences of HBV infection more clearly, then introduce in detail how T cells involve in the mechanism leading to those pathology consequences.

  1. Line 105-114, multiple references are needed to support each of these statements.

  1. Line 187-191, “It is recommended that …”, please clearly state which institutions or authorities recommend this management strategy.

  1. Line 202-207, “Vaccinations are known to be somewhat less effective …”, since the authors recommend that vaccination is the best way to manage this complex situation, the authors should include the details of COVID vaccination. Currently, this paragraph is too vague and has zero reference.

  1. Line 245-248, “The monoclonal antibody tocilizumab (TCZ) is often used in COVID-19 centers to neutralize the negative and life-threatening effects of the cytokines storm. Particularly evaluated in studies on rheumatoid arthritis this drug has induced HBV reactivation 247 nearly 3% of treated patients [78]”. Authors simply points out the situation in rheumatoid arthritis without further discussion. What is the authors’ conclusion or recommendation regarding usage of TCZ in COVID-19 considering its potential effect on HBV reactivation?

Minor issue:

Minor formatting issues should be noted, for example, “SaRS-CoV-2” should be “SARS-CoV-2” in line 26.

Author Response

(x) English language and style are fine/minor spell check required

Answer to the Reviewer: The observation of the reviewer has been accepted and the manuscript has been evaluated by an expert of English language.

In this manuscript titled “Prevention of HBV reactivation in hemato-oncologic setting during COVID-19”, the authors briefly went over the basics of COVID-19 and HBV infections, then introduced the potential HBV reactivation in hemato-oncologic patients, especially in the context of COVID-19 pandemic. Finally, the authors proposed that vaccination against SARS-CoV-2 is the most effective way to deal with such challenging clinical situation. Overall, this is a timely and interesting review. However, I feel that some more detailed information and more references are needed.

 Major issues:

 Point 1: Line 78-79, “There is also nervous system involvement, evidenced by impaired cognitive attention and memory capacity”, references are needed for this statement.

Answer to the Reviewer Point 1: the required references have been added.

Point 2: Line 82, what is the meaning of “pathogenetic mechanism” of HBV infection here? If it means hepatitis, liver failure, cirrhosis, or cancer, it is directly related with hepatocyte death or overgrowth, rather than T cell activation. Please define the pathology consequences of HBV infection more clearly, then introduce in detail how T cells involve in the mechanism leading to those pathology consequences.

Answer to the Reviewer Point 2: the criticisms of the reviewer have been accepted and the manuscript modified accordingly.

Point 3: Line 105-114, multiple references are needed to support each of these statements.

Answer to the Reviewer Point 3: the required references have been added.

Point 4: Line 187-191, “It is recommended that …”, please clearly state which institutions or authorities recommend this management strategy.

 Answer to the Reviewer Point 4: The criticism of the reviewer has been accepted and  institutions and authorities recommending this managing strategies have been clearly stated.

 Point 5: Line 202-207, “Vaccinations are known to be somewhat less effective …”, since the authors recommend that vaccination is the best way to manage this complex situation, the authors should include the details of COVID vaccination. Currently, this paragraph is too vague and has zero reference.

Answer to the Reviewer Point 5: The criticism of the reviewer has been accepted and the manuscript modified accordingly.

Point 6: Line 245-248, “The monoclonal antibody tocilizumab (TCZ) is often used in COVID-19 centers to neutralize the negative and life-threatening effects of the cytokines storm. Particularly evaluated in studies on rheumatoid arthritis this drug has induced HBV reactivation 247 nearly 3% of treated patients [78]”. Authors simply points out the situation in rheumatoid arthritis without further discussion. What is the authors’ conclusion or recommendation regarding usage of TCZ in COVID-19 considering its potential effect on HBV reactivation?

Answer to the Reviewer Point 6: The criticism of the reviewer has been accepted and the manuscript modified accordingly.

Point 7: Minor issue: 

Minor formatting issues should be noted, for example, “SaRS-CoV-2” should be “SARS-CoV-2” in line 26.

Answer to the Reviewer Point 7: The observation of the reviewer has been accepted and the manuscript modified accordingly.

Reviewer 2 Report

This manuscript reviews and analyzes the impact of COVID-19 on onco-hematologic patients, with particular reference to HBV reactivation in HBsAg-positive and HBsAg-negative / HBcAb-positive patients. In general, it is clearly written.

Specific points:

  1. The authors point out that the hepatic inflammation/necrosis results from the attack of HBV-specific cytotoxic T cells targeting viral core antigen on hepatocytes. This is an oversimplification and somewhat incorrect. In addition, it is not viral core antigen but core-derived peptides that are presented on the HBV-infected hepatocytes.
  2. In the first line of the Conlusion, HBsAg-positive should be added.

Author Response

(x) English language and style are fine/minor spell check required

Answer to the Reviewer: The observation of the reviewer has been accepted and the manuscript has been evaluated by an expert of English language.

Point 1: The authors point out that the hepatic inflammation/necrosis results from the attack of HBV-specific cytotoxic T cells targeting viral core antigen on hepatocytes. This is an oversimplification and somewhat incorrect. In addition, it is not viral core antigen but core-derived peptides that are presented on the HBV-infected hepatocytes.

Answer to the Reviewer Point 1: The criticism of the reviewer has been accepted and the manuscript modified accordingly.

Point 2: In the first line of the Conlusion, HBsAg-positive should be added.

Answer to the Reviewer Point 2: The observation of the reviewer has been accepted and the manuscript modified accordingly.

Reviewer 3 Report

Dear authors, I have read your work with interest and attention. Could the administration of the anti covid vaccine by doctors who follow chronically onco-haematological patients lead them to greater adherence to the vaccine itself?
Are there any studies that prove this fact?

Author Response

Point 1: Dear authors, I have read your work with interest and attention. Could the administration of the anti covid vaccine by doctors who follow chronically onco-haematological patients lead them to greater adherence to  the vaccine itself?
Are there any studies that prove this fact?

Answer to the Reviewer Point 1: The observation of the reviewer has been accepted and the manuscript modified accordingly.

We thank the Editor and the Reviewers for helping us to improve our paper.

The manuscript has been read and approved by all the authors.

Round 2

Reviewer 1 Report

All issues resolved.

Author Response

(x) English language and style are fine/minor spell check required

Answer to the Reviewer: The observation of the reviewer has been accepted and the manuscript has been evaluated by an expert of English language.

Reviewer 2 Report

The authors have improved the quality of the manuscript.  Minor spell check is required.

Author Response

(x) English language and style are fine/minor spell check required

The authors have improved the quality of the manuscript.  Minor spell check is required.

Answer to the Reviewer: The observation of the reviewer has been accepted and the manuscript has been evaluated by an expert of English language.